# ACTonFood. Acceptance and Commitment Therapy-Based Group Treatment Compared to Cognitive Behavioral Therapy-Based Group Treatment for Weight Loss Maintenance: An Individually Randomized Group Treatment Trial

**DOI:** 10.3390/ijerph18189558

**Published:** 2021-09-10

**Authors:** Roberto Cattivelli, Anna Guerrini Usubini, Gian Mauro Manzoni, Francesco Vailati Riboni, Giada Pietrabissa, Alessandro Musetti, Christian Franceschini, Giorgia Varallo, Chiara A. M. Spatola, Emanuele Giusti, Gianluca Castelnuovo, Enrico Molinari

**Affiliations:** 1Psychology Research Laboratory, Istituto Auxologico Italiano IRCCS, 20145 Milan, Italy; r.cattivelli@auxologico.it (R.C.); u.guerrini@auxologico.it (A.G.U.); gm.manzoni@auxologico.it (G.M.M.); g.pietrabissa@auxologico.it (G.P.); g.varallo@auxologico.it (G.V.); gianluca.castelnuovo@auxologico.it (G.C.); molinari@auxologico.it (E.M.); 2Department of Psychology, Catholic University of Milan, 20123 Milan, Italy; francesco.vailati@unicatt.it (F.V.R.); chiara.spatola@unicatt.it (C.A.M.S.); 3Faculty of Psychology, eCampus University, 22060 Novedrate, Italy; 4Department of Humanities, Social Sciences and Cultural Industries, University of Parma, 43121 Parma, Italy; alessandro.musetti@unipr.it; 5Department of Medicine and Surgery, University of Parma, 43121 Parma, Italy; christian.franceschini@unipr.it

**Keywords:** obesity, obesity rehabilitation, weight loss maintenance, eating disorders, Acceptance and Commitment Therapy, Cognitive Behavioral Therapy, clinical psychology

## Abstract

The purpose of this Individually Randomized Group Treatment Trial was to compare an Acceptance and Commitment Therapy-based (ACT) group intervention and a Cognitive Behavioral Therapy-based (CBT) group intervention for weight loss maintenance in a sample of adult patients with obesity seeking treatment for weight loss. One hundred and fifty-five adults (BMI: Kg/m^2^ = 43.8 [6.8]) attending a multidisciplinary rehabilitation program for weight loss were randomized into two conditions: ACT and CBT. Demographical, physical, and clinical data were assessed at the beginning of the program (t_0_), at discharge (t_1_), and at 6-month follow-up (t_2_). The following measures were administered: The Acceptance and Action Questionnaire-II (AAQ-II) and the Clinical Outcome in Routine Evaluation-Outcome Measure (CORE-OM). Generalized linear mixed models were performed to assess differences between groups. Moderation effects for gender and Eating Disorders (ED) have been considered. From baseline to discharge, no significant differences between interventions were found, with the only exception of an improvement in the CORE-OM total score and in the CORE-OM subjective wellbeing subscale for those in the CBT condition. From discharge to follow-up, ACT group participants showed significant results in terms of weight loss maintenance, CORE-OM total score, and CORE-OM and AAQ-II wellbeing, symptoms, and psychological problems subscales. Gender moderated the effects of time and intervention on the CORE-OM subscale reporting the risk for self-harm or harm of others. The presence of an eating disorder moderated the effect of time and intervention on the CORE-OM total score, on the CORE-OM symptoms and psychological problems subscales, and on the AAQ-II. Patients who received the ACT intervention were more likely to achieve a ≥5% weight loss from baseline to follow-up and to maintain the weight loss after discharge. The ACT intervention was thus effective in maintaining weight loss over time.

## 1. Introduction

Obesity can be considered as one of the most dominant public-health challenges of the 21st century [1,2,3,4,5,6]. Recent estimates have pointed out how over the last decades obesity has reached epidemic proportions and its prevalence is still rising [7]. Obesity and overweight are frequently associated with many comorbidities. These include type II diabetes mellitus, cardiovascular disease, hypertension, kidney failure, and osteoarthritis [8], as well as psychological problems such as depression, feelings of shame, low self-esteem, stigma, and eating disorders [9,10,11].

Given the complex nature of the phenomenon, multidisciplinary, and multi-component lifestyle interventions aimed at fostering the adoption of a healthier lifestyle through improving healthy eating, increasing physical activity and psychological support, are now highly recommended [12].

Cognitive Behavioral Therapy-based (CBT) treatment is considered the gold standard for the psychological treatment of obesity [12]. Administered in both individual and group settings, this type of intervention aims to enhance self-efficacy, promote coping strategies and problem-solving skills, while addressing dysfunctional cognitions, reinforcing stimulus control, and social activation [10]. CBT for weight management is primarily aimed to foster the individual’s adherence to dietarian and physical activity prescriptions, promoting weight loss, and encouraging the adoption of a healthy lifestyle.

Although CBT has been largely found to be effective in producing weight reduction and consequently reducing the risk for obesity-related health comorbidities [12,13,14], weight loss maintenance still represents an open challenge.

Most individuals with obesity who attended weight loss rehabilitation programs failed to maintain a healthy lifestyle and tended to regain about one-third of the lost weight already over the first following post-treatments year [15,16]. This has spurred worldwide researchers to investigate what factors could play a crucial role in the weight loss maintenance process [17].

Although a significant percentage of weight regain has appeared to be potentially attributable to metabolic efficiency variations, psychological and behavioral factors seem to be better predictors of individual’s adherence to those rehabilitation programs [13,17].

According to Forman’s conceptual model, for example [13], maintaining a healthy lifestyle over time is partially due to some specific self-regulation skills, such as distress tolerance, values clarity, metacognitive awareness, and behavioral commitment. Such skills seem to play a protective role against the tendency to eat palatable food and avoid physical activity in favor of an immediate internal hedonic state of pleasure.

A modern clinical therapeutic approach fostering such self-regulation skills is Acceptance and Commitment Therapy (ACT) [18,19,20,21]. ACT is one of the third waves of CBT aimed to promote psychological flexibility [19]: Described as the ability of “contacting the present moment fully as a conscious human being, and basing on what the situation affords, changing or persisting in behavior in the service of chosen values” [18].

Psychological flexibility results from the interaction of six core therapeutic processes: (a) Getting in contact with what is happening in the present moment, (b) learning to distance from a person’s thoughts, (c) developing a more open attitude to and accepting painful internal experiences, including sensations, emotions, and thoughts, (d) contacting a stable sense of self, regardless of one’s personal experiences, (e) clarifying values, conceptualizing chosen life directions, and (f) pursuing actions or stable behaviors driven by personal values [22].

Despite CBT still being considered as the standard treatment for obesity [12], existing pieces of evidence show promising results supporting the adoption of ACT-based interventions for efficiently addressing weight loss [23,24], and weight maintenance [25]. In “The Mind Your Health Project”, for example [26,27], 190 participants with obesity were randomly assigned to either an ACT-based or CBT-based group intervention for weight loss [28]. Results showed that patients undergoing ACT-based group sessions showed a greater weight loss than those receiving CBT after 12 months of treatment and were more likely to maintain weight loss after a 1-year follow-up. ACT was also found to be significantly effective in addressing comorbid eating disorders [21,22]. By fostering self-regulation skills, ACT could be considered a valid alternative to CBT in promoting the adoption as well as the maintenance of a healthy lifestyle even in those individuals who tend to overeat in response to negative feelings, a phenomenon known as emotional eating, which is common in obesity and eating disorders related to obesity such as Binge Eating Disorder [18,23].

Given these premises, the present study aimed to evaluate the efficacy of a newly developed ACT-based group intervention improving weight-loss maintenance compared with a standard CBT-based group treatment in a sample of Italian participants with obesity involved in a multidisciplinary in-hospital rehabilitation program for weight reduction. Moreover, the present study also aimed to explore treatment differences in terms of psychological inflexibility, experiential avoidance, and psychological treatment outcomes.

No significant differences between the two interventions, in terms of weight loss at discharge, were a-priori hypothesized, due to the same rehabilitation program followed by participants. However, we expected to find greater maintenance of weight loss at the 6-month-follow-up in the ACT-group than in the CBT-condition. To assess the hypothesis that ACT could be suited for patients with eating disorders (i.e., Bulimia Nervosa, Binge Eating Disorder, Eating Disorder Not Otherwise Specified), the moderation effect of eating disorders, as well as gender (as possible confounding variables) were assessed.

## 2. Materials and Methods

### 2.1. Study Design

An individually randomized group treatment trial [29,30] was conducted to compare an ACT-based group intervention and a standard CBT-based group intervention during a multidisciplinary treatment for weight reduction. In the present study design, individuals were randomized into experimental conditions in which treatment was delivered in group form.

### 2.2. Participants/Recruitment of the Study Population, Inclusion, and Exclusion Criteria

Participants were 155 Italian adults with obesity recruited at the IRCCS Istituto Auxologico Italiano—S. Giuseppe Hospital, a clinical center specialized in weight loss interventions located in the North of Italy. In-patients were eligible for the study if they met the following inclusion criteria: (a) Age between 18 and 65 years, (b) BMI (Kg/m^2^) > 30. Written and informed consent to participate were collected for all the participants. Patients were excluded from the study if they had a severe psychiatric diagnosis, other than an eating disorder, or concurrent severe medical conditions potentially compromising study participation. SCID-5-CV was administered before enrollment to check psychiatric comorbidities [31].

### 2.3. Measures

Demographical data were collected through a self-report form. Weight and height were carefully assessed by internal dieticians using standardized procedures to calculate body mass index (BMI, kg/m^2^). Psychological outcomes were collected with the following questionnaires:-The Clinical Outcome in Routine Evaluation-Outcome Measure (CORE-OM) [32]. The Italian validated version [33] is a self-report measure designed to test the psychological treatment outcomes. It is composed of 34 items rated from 0 (never) to 4 (always) on a 5-point Likert scale that covers four domains: Subjective wellbeing (from now on “CORE-wellbeing” 4 items, e.g., “I have felt overwhelmed by my problems”), symptoms/psychological problems (from now on “CORE-Symptoms”, 12 items, e.g., “I have felt unhappy”), life functioning (from now on “CORE-functioning”, 12 items, e.g., “I have felt criticized by other people”) items, and risk for self-harm or harm of others (from now on “CORE-risk”, 6 items, e.g., “I made plans to end my life”), all considered as expressions of distress and dysfunctions [34]. The total score (CORE-total) is the sum of the subscales’ scores. Higher scores reflect worse clinical conditions and psychological distress.-The Acceptance and Action Questionnaire (AAQ-II) [35]. AAQ-II is the most widely used measure of psychological inflexibility and experiential avoidance. We used the validated Italian version of the AAQ-II [36]. It consists of 7 items (e.g., “I am afraid of my feelings”, “I worry about not being able to control my worries and feelings”) rated from 0 (never true) to 7 (always true) on an 8-point Likert scale. Higher scores indicate greater psychological inflexibility, while lower scores are indicators of psychological flexibility. Developed originally as a direct measure of psychological inflexibility, later studies have highlighted how the AAQ-II could be more suited as an indicator of experiential avoidance [37].

### 2.4. Procedures

The study took place at the IRCCS Istituto Auxologico Italiano—S. Giuseppe Hospital, Italy. Patients were selected, in line with this study’s inclusion and exclusion criteria, through a structured clinical interview provided by licensed trained psychologists at the beginning of the program. During the interview, patients were informed about the aims of the study and were asked to sign a written informed consent to participate. Diagnoses of eating disorders were developed in line with established criteria from the Diagnostic and Statistical Manual (DSM-5).

Patients enrolled in the study followed a medical, nutritional, and physical rehabilitation program as usual. Specifically, they adopted a hypocaloric balanced diet provided by the clinic dietitians (energy intake around 80% of the basal energy expenditure estimated, according to the Harris-Benedict equation) (Roza and Shizgal 1984). Moreover, individuals took part in a nutrition counseling program including abomarchut 2 h of individual meeting, integrated with specific educational groups of about 1 h each, focused on nutritional education, information on obesity and its health-related risks, and strategies to manage eating habits. Patients were also involved in physical rehabilitation, performing about 90 min a day of individualized activity for a total estimated volume of about 450 min a week. Each session was a combination of resistance training and endurance, tailored to individual specific needs. All the activities were assisted by trainers, physiotherapists, or other rehabilitation professionals. During the program, individuals also participated in sessions of aerobic activity, walking, or cycling, as well as postural gymnastics, balance training, and other focused activity based on individual needs. The daily caloric expenditure for each patient was estimated at 10% of their basal level.

For the psychological component of the rehabilitation program, individuals were randomly allocated into two different experimental conditions:-ACT-based group intervention. Patients received the standard multidisciplinary rehabilitation program, with the psychological intervention delivered in a group-based setting following the ACT principle.-CBT-based group intervention. Patients received the standard multidisciplinary rehabilitation program with the psychological intervention delivered in a group-based setting following the CBT principle.

Randomization procedures were carried out through a free and open-online service, Randomization.com (http://www.randomization.com accessed on 15 September 2014).

Weight, height, CORE-OM, and AAQ-II were assessed at the beginning of the intervention (Time 0), at the end of the inpatient phase (Time 1), and at the 6-month follow-up from discharge (Time 2). Data were collected during the inpatient phase, with follow-up measures obtained through an online form specifically designed for the current study. The study’s procedure is summarized in Figure 1. Table 1 describes the two interventions.

### 2.5. Intervention

#### 2.5.1. CBT-Based Group Intervention

The CBT-based intervention consisted of three weekly group sessions based on CBT for weight management. Treatment guidelines were developed internally by the authors based on a similar intervention delivered in another context [38] and adapted to the setting conditions, which required brief interventions. The techniques employed were: Healthy lifestyle and emotional eating psychoeducation, goal-setting, imaginal and in vivo (real or imagined) exposures for desensitization to body image triggers, cognitive restructuring, and relapse prevention techniques [39]. Homeworks were assigned at the end of each session and reviewed at the beginning of the next session.

#### 2.5.2. ACT-Based Group Intervention

The ACT-based group intervention consisted of three weekly group sessions based on the ACT framework [25]. The focus of the intervention was to promote psychological flexibility [18]. The treatment’s guidelines were developed by the authors, based on a similar intervention delivered in another context [40], and adapted for the clinical population enrolled in the inpatient obesity rehabilitation program. The interventions were performed according to shared guidelines for administration, maximizing the probability of treatment consistency across groups.

Each session used metaphors and experiential exercises to address the core components of ACT. All sessions included activities and exercises aimed to build ACT skills, for example, mindfulness, acceptance, and values identification. Each participant was supported to clarify personal values and engage in committed actions directly linked to these “freely-chosen values”. Patients were also helped to move from the “conceptualized self” that takes account from defining attributes to the “observing self” who experiences thoughts and feelings but is not defined by any of them. Moreover, patients were helped to make contact with the present moment. They learned to embrace difficult thoughts and emotions through the full range of human experience with kindness and acceptance, rather than attempt to overcome aversive states by eating tasty foods and avoiding physical activity. Finally, they were encouraged to apply defusion strategies, learned through practical experiential exercises aimed to teach how to disentangle oneself from stressful thoughts and feelings to better accept them.

### 2.6. Power Analysis and Sample Size

When power analysis was performed, no other study had compared an ACT-based intervention to a CBT-based intervention for weight loss, psychological flexibility, and psycho-social outcomes in a sample of inpatients with obesity, delivered in a group setting and during hospitalization. Moreover, when the study was planned, no information was available about the possible clustering effects of the ACT-based group intervention. This prevented a regular power analysis calculation for the multilevel models that were planned for data analysis since the a-priori information on the random parts (i.e., between-subject variance and error variance) was not available. Therefore, the sample size was initially calculated for a 2 (between) × 3 (within) ANOVA. We decided, based on both statistical and methodological reasons, to set power = 0.95 since ACT treatments were in the first phases of development and it was necessary to avoid type II errors that could have impaired a proper efficacy evaluation. Setting alpha to 0.05, power to 0.95, the correlation amongst repeated measures to 0.5, and the non-sphericity correction to the lowest value (0.5), a total sample size of 72 participants was needed to detect a medium-size interaction effect (f = 0.25). Taking into account a possible design effect (due to the cluster randomized design) equal to 2 and a probable dropout rate of 10% at follow-up, the number of patients to be recruited was increased to 160 [(72 × 2) + 10%]. The power analysis was performed using the G*Power software (release 3.1.3) [41].

### 2.7. Statistical Analysis

Continuous variables were described using means and standard deviations or using medians and interquartile ranges in the case of non-normal distribution. Categorical variables were described using frequencies and percentages. Missing data were explored using descriptive statistics and graphical methods. To assess if they followed a Missing Completely At Random Mechanism, Little’s MCAR test was performed [42]. Differences between ACT and CBT intervention in terms of baseline demographic and clinical variables were assessed using chi-square tests, Mann-Whitney tests, and t-tests, as appropriate.

To evaluate the ACT vs. CBT efficacy, Generalized Linear Mixed Models (GLMMs) were used. Briefly, these models are a type of multilevel regression analysis that enables the analysis of dependent variables having non-normal distributions. We chose to use multilevel regression models since the ACT and the CBT intervention were administered in groups of 8 patients, and this could represent a violation of the assumption of the independence of the observations of linear regression, and we chose to use GLMM rather than a linear mixed model since the variable CORE-risk was highly skewed [43,44].

GLMMs allow analyzing specific distributions by specifying a link function, i.e., a function that linearizes the relationship between the predictors and the expected values of the dependent variable, and a family distribution, i.e., the expected distribution of the residuals of the model. In the present study, identifying link functions and normal family distributions were used to model changes in BMI, weight, CORE-total, CORE-wellbeing, CORE-symptoms, CORE-functioning, CORE-risk, and AAQ-II. GLMMs conducted using an identity link function and a normal family distribution are considered to be equivalent to a linear mixed model. In contrast, since the CORE-risk was highly right-skewed and most of the patients scored 0 on that scale, a zero-inflated Poisson model with a log link function was employed.

Since GLMMs are multilevel models, they allow assessing random effects, i.e., variables that cluster the data, and fixed effects, i.e., the independent variables. In this study, since the clustering variables were both subject (which clusters the repeated measures of the dependent variables) and group, these variables were assessed as random effects. The random effect of the group variable was dropped, and all the models were re-estimated including only the random intercept for the subjects. Their significance was tested using likelihood ratio tests with critical α set to 0.05 to evaluate if clustering data using these variables was needed in the subsequent models. Fixed effects were time (from baseline to discharge and from discharge to follow-up), intervention (ACT vs. CBT), and their interaction. A significant fixed effect of time suggests that outcome variables change significantly over time, irrespectively of the intervention. A significant effect of intervention suggests the presence of differences between the interventions, irrespective of time. Finally, a significant interaction effect between time and intervention suggests that the two interventions are associated with different courses of the dependent variables over time. Therefore, the interaction effect was interpreted to assess the efficacy of the ACT and the CBT interventions. The significance of the fixed effects was tested by computing 95% Wald confidence intervals of their estimates, in line with well-established procedures.

To build the GLMM models, a bottom-up approach was used. The first model (one for each outcome variable) included only the random intercept for the subjects, the random intercept for the treatment groups, and the fixed effect of time (coded as a categorical variable with three levels: 0 = baseline; 1 = post-treatment; 2 = 6-month follow-up). The effect of time was analyzed utilizing a repeated contrast comparing baseline to discharge and discharge to follow-up. Subsequent models included the fixed effect of the intervention (0 = CBT; 1 = ACT) and its interaction with time. Given the repeated contrast, two interaction effects were estimated: One with the baseline vs. discharge effect and one with the discharge vs. follow-up effect. Simple slopes were estimated to probe significant interactions. Values of the simple slopes can be interpreted as mean changes over time in each intervention group.

Finally, we decided to analyze the moderation effects of gender (0 = female; 1 = male) and ED diagnosis (0 = no ED; 1 = ED). To do so, we added to the proceeding models a fixed effect for each of these variables and the fixed effects of their interaction with time and intervention. A significant three-way interaction between time, intervention, and the moderating variable suggests that the moderating variable influences the changes in the dependent variables differently in the two intervention conditions over time.

All the models were estimated with the Restricted Maximum Likelihood (REML) estimation method. As a sensitivity analysis, all the models were re-estimated using a robust variant of the estimation method [45]. When differences between the REML and robust estimates of the fixed effects were detected and visual inspection of residual plots confirmed the presence of outliers or heteroscedasticity of the residuals, robust results are displayed.

Finally, two logistic regression analyses were performed to assess if patients who received the ACT intervention were more likely than patients who received the CBT intervention to reduce their weight from baseline to follow-up and to maintain the weight loss after discharge. The first logistic regression employed as an outcome a binary variable describing if the patient had achieved a ≥5% weight loss at follow-up compared to the baseline. The second logistic regression employed as an outcome a binary variable describing if the patients had maintained, or decreased, his/her weight from discharge to follow-up. Along with the effect of the intervention, the effect of gender and baseline level of BMI and CORE-total were added in both models to control for confounding. Odds ratio with 95% confidence intervals is reported.

GLMMs were fitted using R (version 3.6.1) packages lme4 (version 1.1.21) and, to analyze the change in CORE-risk scores, glmmTMB [46]. The significance tests of fixed effects and likelihood ratio tests of random effects were performed using the R package lmerTest [47] (version 3.1.1). The R package robustlmm [45] (version 2.2-1) was used for estimating robust models.

### 2.8. Treatment Fidelity

The research group comprises licensed psychologists with previous experience in the field of clinical interventions in health care settings and research. Treatments were delivered by two licensed and experienced clinical psychologists, trained in both approaches and blinded to the final research aims. All the sessions were audiotaped and 20% were randomly chosen to be coded for fidelity. Two bachelor-level observers, also blinded to conditions, received coding training of approximately eight 1-h sessions, checking similar interventions under the supervision of a senior psychologist and ending the training only after achieving 80% of the internal agreement for two subsequent sessions. After the training, they independently coded the randomly selected audio-recorded sessions to evaluate the adherence to the protocol, the coverage of contents, and the use of any additional strategies. They used an internal checklist developed by the authors of the present work detailed for all of the content of each session. Coders had to achieve a minimum of 80% reliability with each other. With a lower level of agreement, the data were discarded [48]. Ratings reflected adherence to the planned discussion topics and overall adherence to the principles of each therapeutic approach.

## 3. Results

One hundred and fifty-five patients were randomized. No patient was lost from baseline to discharge or from discharge to follow-up. The CONSORT diagram of the study is shown in Figure 2. For each variable, percentages of missing values were <5% and their impact was thus considered negligible. The little MCAR test was not significant, suggesting that data followed a Missing Completely at Random mechanism (χ_(74)_ = 93.02, *p* = 0.07).

Descriptive statistics of the baseline characteristics of the sample are shown in Table 2. There were no differences between the ACT and the CBT interventions regarding baseline demographic or clinical variables.

### 3.1. Results of the Generalized Linear Mixed Models

The random effect for the subjects was statistically significant in all the models that included only the fixed effect of time and the random effects for subjects and groups. In contrast, the random intercept for the groups was not statistically significant in any of them (data not reported). This suggested that clustering the subjects based on their groups did not help explain the variability of the outcome variables. Therefore, this random effect was dropped, and all the models were re-estimated including only the random intercept for the subjects.

Except for the analyses of BMI and weight, differences between the REML and robust estimates were detected. Therefore, for the CORE-OM total scores, for its subscales, and for the AAQ-II, the robust estimation method was used.

The treatment factor comparing ACT to CBT and its interactions with the fixed effect of time was then included in all the models. Results of the models evaluating weight, BMI, and CORE-total are reported in Table 3, whereas the results of the models evaluating the CORE-OM subscales and the AAQ-II are reported in Table 4.

Figure 3 shows the changes over time of weight, BMI, and CORE-total, whereas Figure 4 shows the changes of the CORE-OM subscales and of the AAQ-II.

From baseline to discharge, the interaction between treatment and time was significant only for the CORE-total and the CORE-wellbeing subscale. Analysis of simple slopes showed that the CORE-total improved in both groups and that the CORE-wellbeing improved only in the CBT group (Table 5).

From discharge to follow-up, the interaction between time and treatment was significant for weight, BMI, CORE-total, CORE-wellbeing, CORE-symptoms, and AAQ-II. Analysis of simple slopes showed that patients who received the ACT intervention improved from discharge to follow-up regarding weight, BMI, CORE-total, CORE-symptoms, and AAQ, whereas the CORE-wellbeing worsened in patients who received the CBT intervention (see Table 5). No significant interaction between time and treatment was found for the CORE-functioning and CORE-risk subscale.

### 3.2. Moderation Analysis

The interaction effects of gender and ED diagnosis were explored one at a time. Regarding gender, only the three-way interaction effects between this variable, the intervention, and time from baseline to discharge in the CORE-risk were significant (Estimate [95% CI]: −2.84 [−0.42, −1.63]). For male patients in the CBT condition, the CORE-risk scores decreased, whereas for male patients in the ACT condition the CORE-risk increased. For female patients, the CORE-risk scores decreased in both the ACT and CBT conditions from baseline to discharge. No three-way interactions between time from discharge to follow-up, gender, and intervention were significant for any variable.

Significant three-way interactions between time from baseline to discharge, the intervention and the ED diagnosis were found for the CORE-total (estimate [95% CI]: −6.03 [−10.59, −1.46]), CORE-symptoms (estimate [95% CI]: −3.69 [−6.06, −1.31]), and AAQ-II (estimate [95% CI]: −5.06 [−9.90, −0.21]). Regarding the CORE-total and the CORE symptoms, patients without an ED disorder in the CBT condition improved more than patients in the ACT condition. On the contrary, patients with an ED disorder improved more in the ACT condition than in the CBT condition in the same time interval.

Regarding the AAQ-II, patients without an ED disorder receiving the ACT intervention decreased their inflexibility levels, whereas patients in the CBT condition remained stable. In patients with an ED disorder, being in the ACT condition was associated with decreases in psychological inflexibility, whereas being in the CBT condition was associated with an increase in this variable.

No three-way interactions between time from discharge to follow-up, diagnosis, and intervention were significant for any variable.

### 3.3. Results of the Logistic Regression Models

From baseline to discharge, 40 patients (27%) achieved a ≥5% weight loss (28.2% of the patients in the ACT condition and 26.0% of the patients in the CBT condition). At the 6-month follow-up, 75 patients (51%) showed a ≥5% weight loss from baseline (62% of patients in the ACT condition and 40.8% of patients in the CBT condition. Overall, 85 patients (57.8%) maintained or reduced their weight after discharge (64.8% of the patients in the ACT condition and 47.3% of patients in the CBT condition.

After controlling for sex and baseline levels of BMI and CORE-total, patients who received the ACT intervention were more likely to achieve a ≥5% weight loss at follow-up compared to baseline (OR = 2.32, 95% CI = 1.19—4.61). Moreover, they were also more likely to maintain the weight loss from discharge to follow-up (OR = 2.11, 95% CI = 1.08–4.19).

## 4. Discussion

The present study was registered on clinicaltrial.gov. Many relevant interventions that are important to the health care system and, more in general, for health promotion are not subject to regulation by national or international regulatory bodies. For a multidisciplinary intervention grounded in psychological treatments for obesity rehabilitation, adding the registration from an independent agency could potentially help solve ethical issues and improve transparency.

Prospective registration plays a key role to ensure transparency, ensuring deeper methodological evaluation, and helping define at least a draft of the analytic strategies for the data.

As expected, no significant differences from baseline to discharge were found between the two groups, due to the same rehabilitation program they followed, with the only exception for the psychological intervention. Only a significant difference in the treatment outcomes between CBT and ACT from baseline to discharge was found for CORE-total and CORE-wellbeing. CORE-total (patients who attended CBT improved significantly in both subscales). More interestingly, we found that only patients who attended the ACT intervention were able to maintain weight loss after the intervention, as shown in changes in means of weight and BMI scores from discharge to follow-up.

This result was in line with Forman’s “The Mind Your Health Project” [26], where participants who received the ACT-based intervention were more able to maintain weight loss compared to those who received the CBT intervention. The present study, combined with Forman’s studies [26,27], may have the potential to underly a greater ACT treatment effectiveness concerning long-term effects for weight reduction programs.

With regards to the treatment outcomes, CBT was higher than ACT in producing improvement in the clinical outcomes assessed by the CORE-OM from baseline to discharge. Despite these initial improvements, the effect of psychological treatment at follow-up was larger in the ACT condition than in CBT regarding the CORE-OM outcomes.

This difference between the two interventions could be reasonably attributed to the intrinsic different goals of the therapy: While CBT is focused on the reduction of symptoms [49], providing an immediate—but not lasting—relief, ACT, with its mindful components, foster acceptance rather than reduction of suffering, seen as a part of the normal experience of humans [18,50,51,52].

As deeply discussed in the on-topic literature, our study adds more insights on how ACT could potentially be the better treatment option due to its ability to foster self-regulation skills. These abilities may play a crucial role in weight loss maintenance as they seem to affect how individuals react to an everyday stressful life-situation. Accepting pleasure losses rather than coping with emotional distress could therefore represent the main goal for effective rehabilitation programs.

In our study, moderation analyses showed that patients with ED who attended the ACT intervention improved their clinical conditions, as shown by changes in the total score of CORE-OM and reduction of symptoms and psychological problems compared to CBT. These results are consistent with previous evidence in the literature suggesting that an ACT-based approach is particularly suited for patients with ED, who tend to adopt a dysfunctional coping strategy to deal with unpleasant emotions, by overeating.

On the other hand, participants without ED who attended the CBT intervention, improved more in the same domains than patients in ACT did. Again, the focus of CBT on symptoms’ reduction and improvement of problem-solving may explain the greater impact of CBT on patients without ED.

At the present moment, this is the only study comparing ACT and CBT for weight loss that has explored the moderating role of ED, so further studies are still needed.

Finally, concerning psychological flexibility, from baseline to discharge results showed no differences between the two interventions in AAQ-II scores, while from discharge to follow-up, only patients in the ACT condition improved their psychological flexibility, as shown by the decreased scores in AAQ-II. It is worth noting that psychological flexibility is directly targeted only in the ACT condition. Consequently, the larger improvement in the AAQ-II reflects a better focus on the topic, while the change in psychological flexibility for CBT patents is not lasting at mid-term.

The key to the effectiveness of the intervention based on ACT is the promotion of psychological flexibility, a crucial psychological ability for stable behavioral change, to foster the long-term adoption of a healthy lifestyle. Moreover, due to the specific context where the study took place, our results provide additional evidence in supporting the suitability of ACT in health care settings [53,54].

The present work, despite its limitations, strengthens the suitability of ACT intervention for managing a broader lifestyle change with modifications on health-related habits. With only a midterm follow-up we are lacking evidence for a more stable change, but our results suggest the need for a larger and longer terms analysis, starting from the perspective to increase value-oriented health-related change rather than focusing on weight-related goals. The ongoing improvement in weight in the ACT condition seems to suggest more sustainability of value-oriented lifestyle change compared with effective, but less efficient, goal-related CBT intervention, due to the focus on the psychological flexibility process.

Several limitations of the study must be pointed out. First, given that the psychological treatment was provided within the context of a multidisciplinary rehabilitation program, the specific impact of the psychological intervention on the general outcome of treatment is difficult to detect. In addition, the higher specific context of the study makes the results of difficult interpretation and generalization.

The administration of both interventions by the same practitioners could be another limitation. The results may have been influenced by the therapist’s background, or by his or her preference or increased competence in either approach. Another limitation concerns the adoption of only self-report measures that could suffer from biases and limitations. Finally, a follow-up of 6 months might be inadequate to evaluate the maintenance of behavioral change over time, even if it is generally considered adequate as a mid-term outcome [55].

Despite these limitations, this study could represent an important advancement in the field of obesity management, addressing the problem of providing effective interventions able to promote the adoption of lasting healthy behaviors and psychological wellbeing in individuals with obesity.

ACT protocols may represent a feasible approach that could be easily replicated in different primary care settings, offering individuals at the same time functional mindful skills applicable to everyday situations. Individuals trained with ACT principles may therefore potentially observe a more long-lasting effect, crucial for loss weight maintenance.

Future studies are necessary to better clarify the mechanisms of changes and which psychological factors may impact weight maintenance for different individuals. In addition, future replications of the study should take under consideration a longer period to set follow-up measures and a larger number of measures assessed.

## 5. Conclusions

Obesity may be considered as one dominant public-health challenge of our century [1,2,3,4,5,6]. Moreover, weight-loss maintenance still represents one key variable to be investigated deeper by researchers.

A modern clinical therapeutic approach offering promising results is represented by the Acceptance and Commitment Therapy (ACT) [18]. The modern literature offers significant evidence supporting the adoption of ACT-based interventions for efficiently addressing weight loss [23,24] and weight maintenance. In the present study, we evaluated the efficacy of an ACT-based group intervention improving weight-loss maintenance and enhancing clinical conditions compared with a standard CBT-based group treatment in a sample of Italian participants involved in a multidisciplinary in-hospital rehabilitation program for weight reduction. The results achieved, in the present work, provide evidence suggesting that ACT could be a valid alternative to the gold standard for the psychological treatment of obesity that, as discussed above, fails to achieve long-term weight loss [6].

## Figures and Tables

**Figure 1 ijerph-18-09558-f001:**
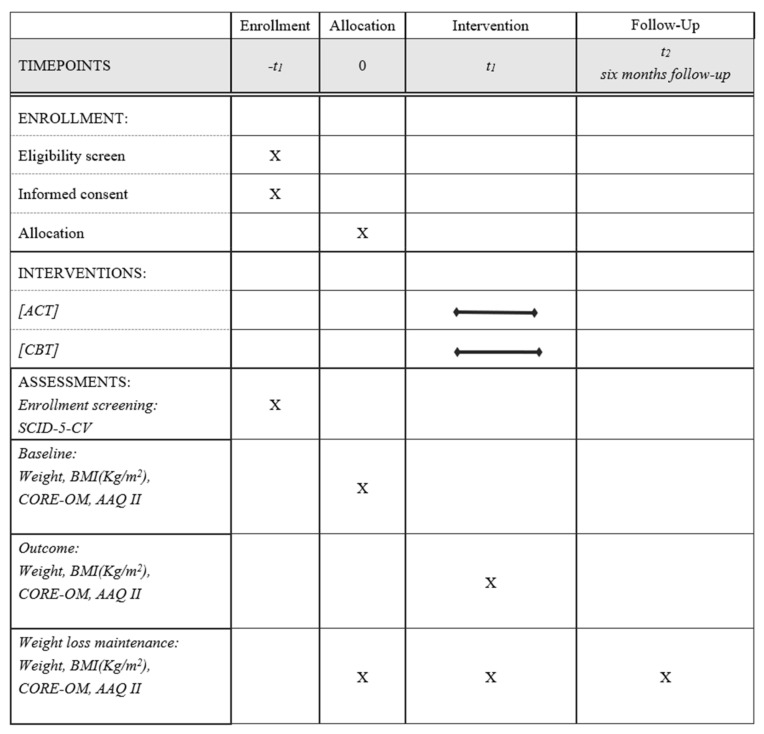
SPIRIT checklist illustrating the study procedures.

**Figure 2 ijerph-18-09558-f002:**
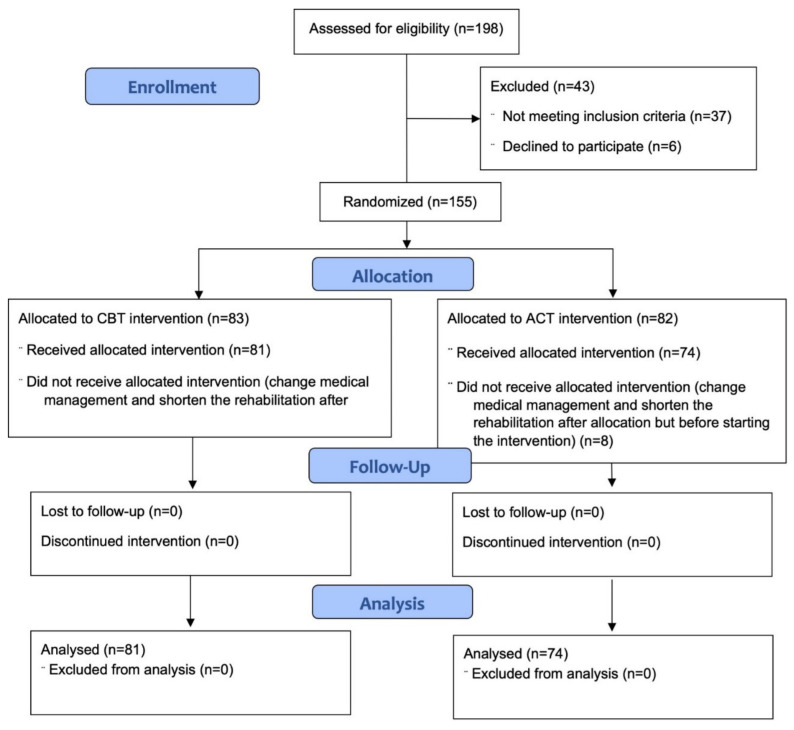
CONSORT diagram illustrating the enrollment procedures.

**Figure 3 ijerph-18-09558-f003:**
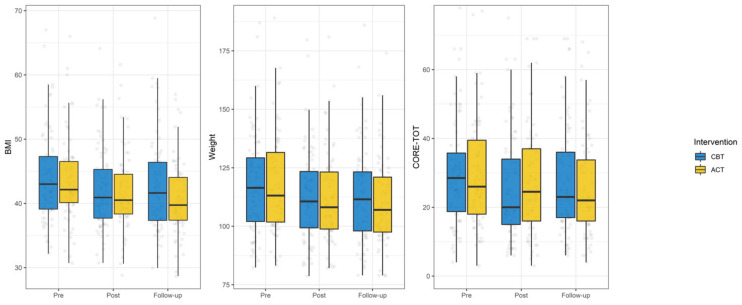
Changes over time in BMI, weight, and CORE-total.

**Figure 4 ijerph-18-09558-f004:**
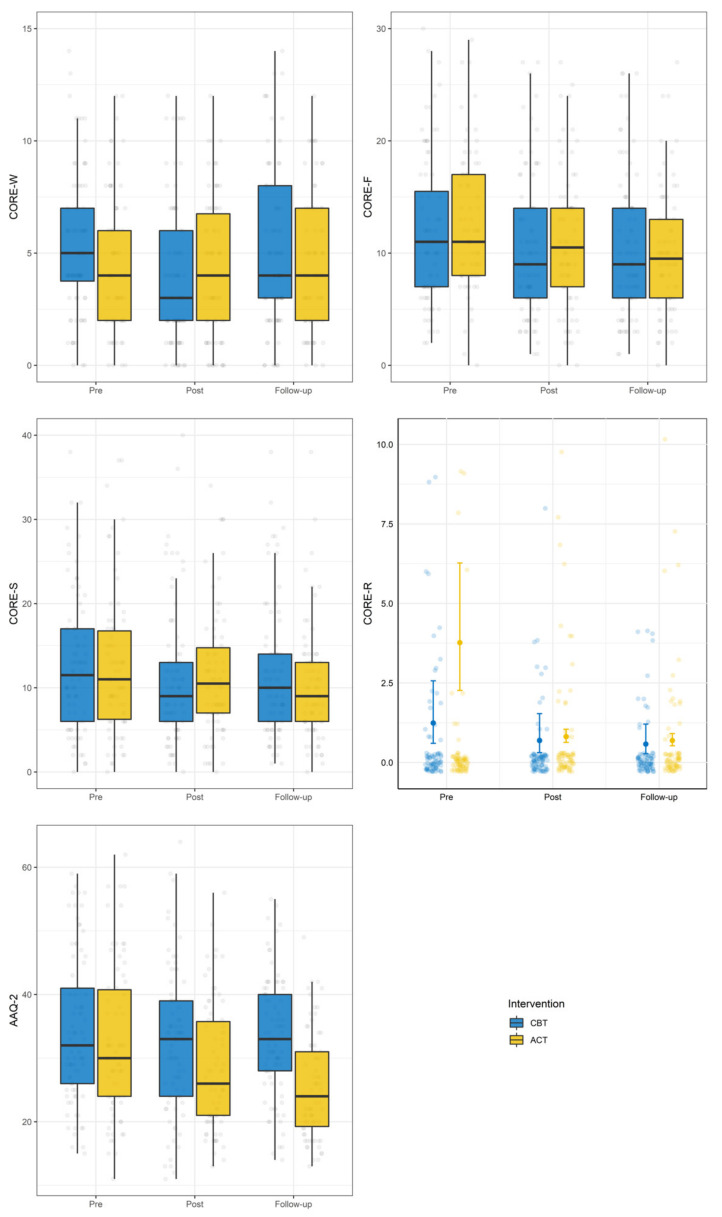
Changes over time of CORE-OM subscales and AAQ-II.

**Table 1 ijerph-18-09558-t001:** Contents of ACT and CBT group treatment.

	ACT	CBT
Session 1	Introduction to the program Group set up Group session overview Root metaphor—Passenger in the Bus Introducing Values Values and obstacles to value-based actions Choice point exercise Summary of the session Proposal of session activity exercise—Bull’s Eye exercise [Value-based Actions]	Introduction to the program Group set up Group session overview Obesity Health Problem PsychoEducation Eating, Food, and Habits Emotional Eating Triggers—Imaginative Exposure Summary of the session Proposal of session activity exercise—food journal
Session 2	Group set up Group session overview Review of session activity exercise—Bull’s Eye [Value-based Actions] Revisit ROOT metaphor—Passengers on the bus Introducing Awareness Mindfulness of body and breath exercise Noticing others’ values—perspective-taking exercise Summary of the session Proposal of session activity exercise—Mindful eating exercise	Group set up Review of session activity exercise—food journal Group session overview Rehabilitation Goal Setting Revisiting: PsychoEducation Emotional Eating Triggers Cognitive Restructuring Summary of the session
Session 3	Group set up Group session overview. Review of session activity exercise—Mindfulness eating Introducing De-Fusion and Willingness (Acceptance) Clouds in the sky mindfulness/defusion exercise Sticky labels defusion exercise Pragmatic Planning—activity exercise Revisit ROOT metaphor—Passengers on the bus Summary of the program Questions and Answers and contacts for future interactions	Group set up Group session overview Review of session activity exercise—Plan to cope with Emotional Eating Triggers Relapse Prevention Summary of the program Proposal of session activity exercise—Relapse prevention plan

**Table 2 ijerph-18-09558-t002:** Descriptive statistics of the total sample and of the ACT and CBT groups at baseline.

		Total Sample (*n* = 155)	CBT (*n* = 81)	ACT (*n* = 74)	*p*-Value **
Gender	Male	112 (72.3)	63 (77.8%)	49 (66.2%)	0.15
	Female	43 (27.7)	18 (22.2%)	25 (33.8%)	
ED diagnosis	None	98 (63.2)	49 (60.5%)	49 (66.2%)	0.69
	NOS	13 (10.4)	8 (9.9%)	5 (6.8%)	
	BED	44	24 (29.6%)	20 (27.0%)	
Weight		117.6 (21.1)	117.41(21.18)	117.89 (21.13)	0.89
BMI		43.8 (6.8)	44.10 (6.99)	43.57 (6.43)	0.63
CORE-total		33.3 (11.7)	30.39 (16.45)	30.07 (16.46)	0.90
CORE-wellbeing		4.9 (3.1)	5.30 (3.11)	4.46 (3.02)	0.09
CORE-symptoms		12.6 (8.4)	12.61 (8.23)	12.54 (8.55)	0.96
CORE-functioning		12.2 (6.3)	11.81 (6.64)	12.53 (5.98)	0.48
CORE-risk		0 [0, 0]	0.78 (1.84)	0.53 (1.86)	0.07
AAQ-II		30.2 (16.4)	34.38 (11.49)	32.22 (11.85)	0.24

Note. Categorical variables are reported with frequencies. Continuous variables are reported with means (SD). Abbreviations: ACT: Acceptance and Commitment Therapy; CBT: Cognitive Behavioral Therapy; ED: Eating Disorders; NOS: Not Otherwise Specified; BED; Binge Eating Disorder; BMI: Body Mass Index; CORE-wellbeing: Clinical Outcome in Routine Evaluation-Wellbeing; CORE-symptoms: Clinical Outcome in Routine Evaluation-Symptoms/Problems; CORE-functioning: Clinical Outcome in Routine Evaluation-functioning; CORE-risk: Clinical Outcome in Routine Evaluation-Risk; CORE-total: Clinical Outcome in Routine Evaluation-total score; AAQ-II: Acceptance and Action Questionnaire-II. ** *p*-value based on chi-square tests for categorical variables, Mann-Whitney test for continuous non-normal variables, and *t*-tests for continuous variables.

**Table 3 ijerph-18-09558-t003:** Results of the generalized linear mixed models comparing the effects of the ACT and CBT treatments overtime on BMI, weight, and on the total score of the CORE-OM.

	BMI		Weight		CORE-Total	
	Est.	95% CI	Est.	95% CI	Est.	95% CI
Intercept	42.33	[41, 43.66] *	113.03	[108.63, 117.43] *	26.34	[23.18, 29.5] *
Time T0–T1	−1.79	[−2.19, −1.39] *	−4.66	[−5.73, −3.59] *	−3.96	[−5.46, −2.45] *
Time T1–T2	0.36	[−0.04, 0.76]	0.68	[−0.4, 1.75]	1.73	[0.23, 3.23] *
Intervention	−0.70	[−2.61, 1.21]	−0.70	[−7.07, 5.67]	−0.08	[−4.65, 4.49]
Time T0–T1*Intervention	0.05	[−0.52, 0.62]	−0.49	[−2.03, 1.05]	2.60	[0.42, 4.77] *
Time T1–T2*Intervention	−1.31	[−1.89, −0.74] *	−2.84	[−4.39, −1.3] *	−3.76	[−5.93, −1.6] *

Note. Coefficients are unstandardized. The fixed effects “Time T0-T1” and “Time T1-T2” indicate a change in the outcomes from baseline to discharge and from discharge to follow-up, irrespectively of intervention. The fixed effect “Intervention” indicates differences between the ACT and CBT interventions, irrespectively of time. The interaction terms “Time T0–T1*Intervention” and “Time T1–T2*Intervention” indicate changes over time of the outcomes due to having received the ACT or the CBT intervention. Abbreviations: ACT: Acceptance and Commitment Therapy; CBT: Cognitive Behavioral Therapy; BMI: Body Mass Index; CORE-total: Clinical Outcome in Routine Evaluation-total score. * Significant effect.

**Table 4 ijerph-18-09558-t004:** Results of the generalized linear mixed models comparing the effects of the ACT and CBT treatments over time on the subscales of the CORE-OM and on the AAQ-II.

	CORE-Wellbeing		CORE-Symptoms		CORE-Functioning		CORE-Risk		AAQ-II	
	Est.	95% CI	Est.	95% CI	Est.	95% CI	Est. ^a^	95% CI	Est.	95% CI
Intercept	4.58	[3.9, 5.27] *	10.72	[9.29, 12.15] *	10.48	[9.19, 11.77] *	−0.99	[−1.85, −0.12] *	32.86	[30.71, 35.01] *
Time T0–T1	−1.42	[−1.85, −0.99] *	−1.51	[−2.31, −0.72] *	−1.34	[−2.08, −0.59] *	0.50	[−1.56, 2.56]	−2.00	[−3.61, −0.4] *
Time T1–T2	1.17	[0.75, 1.6] *	0.51	[−0.29, 1.3]	−0.05	[−0.79, 0.7]	−0.13	[−2.05, 1.78]	0.86	[−0.74, 2.47]
Intervention	−0.22	[−1.21, 0.78]	−0.22	[−2.3, 1.85]	0.41	[−1.46, 2.28]	−0.08	[−1.15, 1]	−4.52	[−7.63, −1.41] *
Time T0–T1*Intervention	1.33	[0.71, 1.95] *	1.06	[−0.09, 2.21]	−0.28	[−1.36, 0.79]	0.44	[−0.36, 1.24]	−1.66	[−3.98, 0.67]
Time T1–T2*Intervention	−0.91	[−1.52, −0.29] *	−1.84	[−2.98, −0.69] *	−0.79	[−1.86, 0.29]	−0.05	[−0.7, 0.61]	−3.59	[−5.92, −1.27] *

Note. Coefficients are unstandardized. The fixed effects “Time T0-T1” and “Time T1-T2” indicate change in the outcomes from baseline to discharge and from discharge to follow-up, irrespectively of intervention. The fixed effect “Intervention” indicates differences between the ACT and CBT interventions, irrespectively of time. The interaction terms “Time T0–T1*Intervention” and “Time T1–T2*Intervention” indicate changes over time of the outcomes due to having received the ACT or the CBT intervention. Abbreviations: ACT: Acceptance and Commitment Therapy; CBT: Cognitive Behavioral Therapy; ED: CORE-wellbeing: Clinical Outcome in Routine Evaluation-Wellbeing; CORE-symptoms: Clinical Outcome in Routine Evaluation-Symptoms/Problems; CORE-functioning: Clinical Outcome in Routine Evaluation-Functioning; CORE-risk: Clinical Outcome in Routine Evaluation-Risk; AAQ-II: Acceptance and Action Questionnaire-II. * Significant effect. ^a^ Estimates based on a zero-inflated Poisson model with a log link function.

**Table 5 ijerph-18-09558-t005:** Results of the simple slope analysis performed to assess differences between ACT and CBT in changes of the outcomes over time.

	ACT			CBT		
	Coefficient	SE	*p*-Value	Coefficient	SE	*p*-Value
From baseline to discharge						
CORE-total	−2.01	1.00	0.04	−4.39	0.96	<0.01
CORE-wellbeing	−0.08	0.29	0.89	−1.25	0.28	<0.01
From discharge to follow-up						
Weight	−2.31	0.61	<0.01	0.65	0.59	0.27
BMI	−0.96	0.22	<0.01	0.41	0.21	0.05
CORE-total	−2.18	1.00	0.03	1.77	0.96	0.06
CORE-wellbeing	0.28	0.30	0.33	1.34	0.28	<0.01
CORE-symptoms	−1.45	0.54	<0.01	0.51	0.51	0.32
AAQ-II	−2.92	0.94	<0.01	0.83	0.90	0.36

Note. The simple slope analyses were performed when the interaction between time and treatment was significant. Values of the coefficients can be interpreted as mean changes from baseline to discharge and from discharge to follow-up across the interventions. Coefficients are unstandardized. Abbreviations: ACT: Acceptance and Commitment Therapy; CBT: Cognitive Behavioral Therapy; CORE-total: Clinical Outcome in Routine Evaluation-total score; CORE-wellbeing: Clinical Outcome in Routine Evaluation-Wellbeing; CORE-symptoms: Clinical Outcome in Routine Evaluation-Symptoms/Problems; AAQ-II: Acceptance and Action Questionnaire-II.

## Data Availability

The data presented in this study are available on request from the corresponding author. The data are not publicity available due to participants did not accept their data to be share with third bodies or people besides the research team.

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
