# Peer review of "ACTonFood. Acceptance and Commitment Therapy-Based Group Treatment Compared to Cognitive Behavioral Therapy-Based Group Treatment for Weight Loss Maintenance: An Individually Randomized Group Treatment Trial"

_ijerph, 2021, doi:10.3390/ijerph18189558_

Round 1

Reviewer 1 Report

“ACTonFood. Acceptance and Commitment Therapy-based 2 group treatment compared to Cognitive Behavioral Therapy based group treatment for weight maintenance: an individually randomized group treatment trial”.

Thank you for inviting me to review this RCT comparing two active treatment groups for weight management (CBT vs. ACT).  The trial design is generally sound but changes are required to the manuscript to bring it up to a publishable standard. For example, a CONSORT flowchart is missing and this will ensure that all information on the reporting of a trial is included.  I am also concerned that the authors state that their trial registration number is NCT01096251 but upon searching clinicaltrials.gov, this appears to be a different, already reported study from the same group. The authors need to provide the correct trial number so that the reviewers can compare what they stated they would do with what they are reporting they did in this article. Once this has been provided, a further review can be undertaken.

Some other specific points that need addressing are below:

INTRODUCTION

Line 70 – you insert a paragraph here about Forman’s conceptual model but it could be better linked to the paragraph before this to improve the flow.

Line 76 – “Raising in the last twenty years” needs editing as grammatically doesn’t make sense.

Line 86 – “Despite CBT is still nowadays considered as the 86 treatment of choice for obesity” needs editing as grammatically doesn’t make sense.

Towards the end of your introduction, you reference a number of trials of ACT vs CBT and so it would be helpful to be explicit about why this current trial needed to be complete if there had been success already?  You make references to differences in the power analysis section but this is the first time these are referenced explicitly

You mention ‘eating disorders’ towards the end. Could you be a bit more explicit about what you are referring to from a diagnostic perspective e.g. Bulimia nervosa?

METHODS

As this was a trial, please present what your primary outcome was.

CONSORT checklist for the reporting of trials is also missing

Blinding – who undertook the outcome assessments and were they blind to which group the participant was in?

How did you diagnose eating disorders – did you undertake a SCID assessment for example? This should be included in your baseline measures.

RESULTS

The numbers in the CBT group do not add up. In the ACT group the number isn’t visible in the CONSORT diagram.

Results should start with presenting the primary outcome.

It would be more meaningful to readers to be able to see mean changes (and 95% CIs) between the time points as part of the results, this provision of raw data would also be helpful for potential inclusion in future systematic reviews

DISCUSSION

It would be good to reference the previous trials you mention in your introduction near the beginning of your discussion and how these results compare to your current findings

Line 465 – please clarify what you mean by “Despite these initial improvements, the effect of psychological treatment at follow up was larger in the ACT condition than in CBT.” What are you measuring this ‘effect’ as?

Line 514 needs to be deleted

Author Response

Response to reviewer comments for:

- Manuscript ID: ijerph-1305537

- Title: “ACTonFood. Acceptance and Commitment Therapy-based group treatment compared to Cognitive Behavioral Therapy-based group treatment for weight maintenance: an individually randomized group treatment trial”

- Journal: International Journal of Environmental Research and Public Health

We genuinely thank all reviewers for the time dedicated to revising our paper, their interest in our study, and all of their meaningful suggestions.

Provided below is a point-by-point response describing our attempts to address all of their revisions in our manuscript.

REQUESTED REVISION

Reviewer 1

  • the authors state that their trial registration number is NCT01096251 but upon searching clinicaltrials.gov, this appears to be a different, already reported study from the same group. The authors need to provide the correct trial number so that the reviewers can compare what they stated they would do with what they are reporting they did in this article.

The correct trial registration number has been inserted in the paper in line with the reviewer’s comment.

INTRODUCTION

  • Line 70 – you insert a paragraph here about Forman’s conceptual model but it could be better linked to the paragraph before this to improve the flow.

Thanks to the reviewer's suggestion, the entire section has been revised and the specific part is better linked to the rest of the paragraph to improve the article’s flow.

  • Line 76 – “Raising in the last twenty years” needs editing as grammatically doesn’t make sense.

In line with the reviewer's suggestion, the line has been rephrased.

  • Line 86 – “Despite CBT is still nowadays considered as the 86 treatment of choice for obesity” needs editing as grammatically doesn’t make sense.

The entire section has been revised thanks to the reviewers' comments. The highlighted line has been changed to increase comprehension.

  • Towards the end of your introduction, you reference a number of trials of ACT vs CBT and so it would be helpful to be explicit about why this current trial needed to be complete if there had been success already?  You make references to differences in the power analysis section but this is the first time these are referenced explicitly

The entire section has been revised thanks to reviewers' comments and suggestions. New explicit statements have been added to the text to briefly explain the different goals of our present study compared to those of the studies already published. We believe that the exploration of treatment differences in terms of psychological inflexibility, experiential avoidance, and psychological treatment outcomes discussed in our present study may potentially expand the on-topic knowledge of ACT treatments Vs CBT treatments.

  • You mention ‘eating disorders’ towards the end. Could you be a bit more explicit about what you are referring to from a diagnostic perspective e.g. Bulimia nervosa?

Thanks to the reviewer's suggestion the text have been revised and specific eating disorders included in the section to facilitate readers' understanding of the current paper.

METHODS

  • As this was a trial, please present what your primary outcome was.

Thanks to the reviewer's suggestion the entire section has been revised by the authors. Our selected primary outcome for the current study has been the psychological clinical outcome, assessed with the CORE-OM questionnaire. However, mid-term weight maintenance and AAQ-II scores were also considered as meaningful outcomes in the present paper.

  • blinding – who undertook the outcome assessments and were they blind to which group the participant was in?

Thanks to reviewer 1 for the above comment enabling the authors to clarify this crucial aspect of the study. Outcomes assessments have been undertaken by different clinical figures working at the Institute Pian Cavallo. Weight for example has been assessed by instructed dieticians blind to which group individuals were enrolled in. Psychological assessment was conducted by the rest of the psychologists equip not involved in the present study and therefore blind to which group individuals were enrolled in.

  • How did you diagnose eating disorders – did you undertake a SCID assessment for example? This should be included in your baseline measures.

In line with the reviewer's suggestion, the section has been revised and specific references to the SCID-5 were added in the text. SCID-5 was administered before enrolment to check psychiatric comorbidities.

RESULTS

  • The numbers in the CBT group do not add up. In the ACT group the number isn’t visible in the CONSORT diagram.

Thanks to the reviewer comment the numbers have been re-checked and fixed. The Figure has been eliminated and a new figure with all visible details is now inserted in the present paper.

  • Results should start with presenting the primary outcome.

In line with the reviewer's suggestion, the entire section has been revised, and now the text flow follows a more logical sequence.

  • It would be more meaningful to readers to be able to see mean changes (and 95% CIs) between the time points as part of the results, this provision of raw data would also be helpful for potential inclusion in future systematic reviews

Tables have been revised in line with reviewer suggestions. We believe the new format could increase clarity for potential readers.

DISCUSSION

  • It would be good to reference the previous trials you mention in your introduction near the beginning of your discussion and how these results compare to your current findings

In line with the reviewer's comment, references to the previous study have been mentioned at the beginning of the discussion section. The entire section has been revised and consideration and comparisons between the different studies added to the present text.

  • Line 465 – please clarify what you mean by “Despite these initial improvements, the effect of psychological treatment at follow up was larger in the ACT condition than in CBT.” What are you measuring this ‘effect’ as?

Thanks to the reviewer's suggestion the entire section has been revised. The paragraph has been rephrased to add more clarity. We believe the current revised section could potentially offer more clarity to the paper.

Reviewer 2 Report

Thank you for the opportunity to review this manuscript outlining weight loss maintenance for two interventions (CBT and ACT-based group treatment) among individuals with obesity who were inpatients in a weight loss rehabilitation program in Italy. The findings of this study have the potential to shape weight loss interventions, both in primary care settings and potentially also for community-based real-world settings. I commend the authors on this novel study, and feel that with some improvements it is worthy of publication.

While the manuscript is mostly easy to understand, it would be worth having it proofread to ensure appropriate wording and use of English language throughout. Although this is an editorial decision, most journals prefer the use of ‘people first’ language (individuals with overweight and obesity, rather than overweight and obese individuals) – this is used in places but not consistently throughout the manuscript. Some suggestions for improvements to the manuscript are outlined below, with particular attention required to improving the Discussion.

  1. TITLE and KEYWORDS

The title clearly reflects the focus of the study, with the exception of the use of the term ‘weight maintenance’. I suggest that the authors change this to weight loss maintenance which more accurately reflects what was assessed in this study. Weight maintenance is a different concept to weight loss maintenance, so suggest also changing the key words to weight loss maintenance instead of weight maintenance.

  1. ABSTRACT

Page 1, Line 24: The abstract refers to the participants as overweight, but the BMI is in the range for obesity, which should be reflected as such in the abstract.

  1. INTRODUCTION

The introduction is well grounded in the literature and provides a good overview of CBT and ACT and their relationship to weight loss. However, the flow of ideas could be improved. The first and third paragraphs are very long, and the second paragraph is very short. Suggest breaking up the introduction more by including more paragraphs with grouping of concepts which may improve the flow of this section.

Page 2, Lines 53, 54: the text mentions nutrition and dieting assessments as part of lifestyle interventions. This is confusing as an assessment is generally not part of an intervention. The sentence is also long and difficult to read. The authors may consider re-wording to: Given the complex nature of the phenomenon, multidisciplinary and multi-component lifestyle interventions aimed at fostering the adoption of a healthier lifestyle through improving healthy eating, increasing physical activity and psychological support, are now highly recommended.

Page 2, line 55: the first paragraph of the introduction is long. Suggest starting a new paragraph from the sentence beginning CBT based treatment is considered the gold standard…

Page 2, line 64: Loss-weight maintenance should be re-worded to weight loss maintenance.

Page 3, Line 101-102: it is not clear what ‘clinical conditions’ are. This needs to be explained more clearly for a number of reasons: to make it clear to the reader exactly what is the outcome of interest is, and to set that in the context of what has been presented in the introduction (i.e.. what is the justification for assessing ‘clinical conditions’?). My apologies if I have misunderstood your intention, but clinical conditions could mean a range of ‘conditions’/

  1. METHODS

Page 3, Line 120-121: I am unfamiliar with the Italian health context, but ‘weight loss procedures’ infers that something is ‘done’ to the patient (e.g. bariatric surgery). The authors may consider using different wording (e.g. weight loss interventions) which is broader and reflects the types of weight loss approaches reflected in this study.

Page 3, Line 124-125: Authors should consider re-wording severe psychiatric disturbance different from eating disorders (suggest – if they had a severe psychiatric diagnosis, other than an eating disorder …).

Page 3, Line 129: weight and height were ‘carefully assessed’ – who did the weight and height measurements?

Page 3, Line 130: Other clinical measures (what does this mean?). Suggest using a more descriptive term here which outlines what is to be measured.

Page 3, Line 147: Check rating scale. 0-7 is an 8-point scale, not a 7-point scale.

Page 4, Line 153-155: Suggest simplifying this sentence as the setting has previously been described in the manuscript.

Page 5, Line 191 (Figure 1): Assessments – the table outlines three rows (baseline, outcome, weight loss maintenance). The outcomes measured are listed (but I am not sure how the bolded ‘baseline’, ‘outcomes’ and ‘weight loss maintenance’ relate to the measures. This needs to be clearer. It is also confusing that under weight loss maintenance, physical activity is measured, but not weight or BMI?

Page 5, Lines 198-199: I do not know what ‘imaginal and in vivo exposures’ are (is it possible to include a brief description?).

Page 5, Line 205-207: This quote is included in the introduction and need not be repeated here.

Page 5, Line 207: I do not know what ‘manualised internally by the authors’ means – please use simpler language.

Page 6, Table 1: This is a good table which outlines each intervention well. It would be good to more clearly differentiate between Sessions 1,2 and 3, either by including a line between them, or using shading.

  1. RESULTS

Page 9, Figure 2: a) the last line in the allocation cell for ACT is missing. I assume it should include n=8?

  1. b) For the CBT allocation cell, the authors state that n=82 received intervention but 81 were allocated to intervention and 2 did not receive intervention (adds up to 83). Check numbers.

Page 13, Lines 443-446: More than half of the participants continued to lose weight during the follow up period – more could be made of this in the discussion.

  1. DISCUSSION

Overall, the focus of the discussion seems to be heavily on the scores of CORE-OM and AAQ-II which is all relevant (as long as it is not simply repeating the results). The implications of these findings needs to be more specifically discussed, with reference to the literature and as this is a novel study, borrowing from other areas where such interventions have been successfully implemented could be included.

The discussion lacks mention of how these findings fit with current literature and thinking around weight loss maintenance interventions, and in particular how the behaviour change literature in primary care settings. It would also be of interest to readers to set this more in the context of how ACT and its focus assists participants to maintain weight loss once they are no longer in an in-patient setting. What is the real-world relevance of this research? And how do psychological interventions such as these fit within the context of lifestyle behaviour change interventions.

  1. CONCLUSION

Page 15, Lines 519-522: This paragraph is a repeat of what is included in the introduction. Suggest simplifying and re-wording.

Author Response

Reviewer 2

  1. TITLE and KEYWORDS
  • The title clearly reflects the focus of the study, with the exception of the use of the term ‘weight maintenance’. I suggest that the authors change this to weight loss maintenance which more accurately reflects what was assessed in this study. Weight maintenance is a different concept to weight loss maintenance, so suggest also changing the key words to weight loss maintenance instead of weight maintenance.

Title and keywords have been revised and changed thanks to reviewer suggestions. We believe this new version could potentially offer more clarity to readers.

  1. ABSTRACT
  • Page 1, Line 24: The abstract refers to the participants as overweight, but the BMI is in the range for obesity, which should be reflected as such in the abstract.

Thanks to the reviewer's comment the entire abstract has been revised according to suggestions.

  1. INTRODUCTION
  • The introduction is well grounded in the literature and provides a good overview of CBT and ACT and their relationship to weight loss. However, the flow of ideas could be improved. The first and third paragraphs are very long, and the second paragraph is very short. Suggest breaking up the introduction more by including more paragraphs with grouping of concepts which may improve the flow of this section.

In line with reviewer suggestions, the entire introduction section has been revised and broken up into different paragraphs to improve the flow.

  • Page 2, Lines 53, 54: the text mentions nutrition and dieting assessments as part of lifestyle interventions. This is confusing as an assessment is generally not part of an intervention. The sentence is also long and difficult to read. The authors may consider re-wording to: Given the complex nature of the phenomenon, multidisciplinary and multi-component lifestyle interventions aimed at fostering the adoption of a healthier lifestyle through improving healthy eating, increasing physical activity and psychological support, are now highly recommended.

In line with the reviewer's comment, the text has been rephrased accordingly.

  • Page 2, line 55: the first paragraph of the introduction is long. Suggest starting a new paragraph from the sentence beginning CBT based treatment is considered the gold standard…

The first paragraph has been divided into different sub-paragraphs in line with reviewer suggestions.

  • Page 2, line 64: Loss-weight maintenance should be re-worded to weight loss maintenance.

The phrase has been revised and re-worded according to the reviewer's comment.

  • Page 3, Line 101-102: it is not clear what ‘clinical conditions’ are. This needs to be explained more clearly for a number of reasons: to make it clear to the reader exactly what is the outcome of interest is, and to set that in the context of what has been presented in the introduction (i.e.. what is the justification for assessing ‘clinical conditions’?). My apologies if I have misunderstood your intention, but clinical conditions could mean a range of ‘conditions’/

Thanks to reviewer suggestions, the entire section has been revised by the authors. More specific explanations have been added to the text to increase clarity.

  1. METHODS
  • Page 3, Line 120-121: I am unfamiliar with the Italian health context, but ‘weight loss procedures’ infers that something is ‘done’ to the patient (e.g. bariatric surgery). The authors may consider using different wording (e.g. weight loss interventions) which is broader and reflects the types of weight loss approaches reflected in this study.

According to reviewer concerns, the entire section has been revised and different wording inserted to increase clarity

  • Page 3, Line 124-125: Authors should consider re-wording severe psychiatric disturbance different from eating disorders (suggest – if they had a severe psychiatric diagnosis, other than an eating disorder …).

The phrase has been re-worded in line with the reviewer's comment.

  • Page 3, Line 129: weight and height were ‘carefully assessed’ – who did the weight and height measurements?

Specific detailed information has been added to the section to clarify assessments.

  • Page 3, Line 130: Other clinical measures (what does this mean?). Suggest using a more descriptive term here which outlines what is to be measured.

In line with the reviewer's suggestion, the entire phrase has been reworded with more descriptive terms.

  • Page 3, Line 147: Check rating scale. 0-7 is an 8-point scale, not a 7-point scale.

Thanks to the reviewer comment the rating scale has been now carefully revised and revised.

  • Page 4, Line 153-155: Suggest simplifying this sentence as the setting has previously been described in the manuscript.

The specific sentence has been revised and simplified according to the reviewer's comment.

  • Page 5, Line 191 (Figure 1): Assessments – the table outlines three rows (baseline, outcome, weight loss maintenance). The outcomes measured are listed (but I am not sure how the bolded ‘baseline’, ‘outcomes’ and ‘weight loss maintenance’ relate to the measures. This needs to be clearer. It is also confusing that under weight loss maintenance, physical activity is measured, but not weight or BMI?

The figure has been revised according to the reviewer's comment.

  • Page 5, Lines 198-199: I do not know what ‘imaginal and in vivo exposures’ are (is it possible to include a brief description?).

A brief description has been included in the section to increase clarity in line with the reviewer's suggestion.

  • Page 5, Line 205-207: This quote is included in the introduction and need not be repeated here.

Quote eliminated from the text thanks to reviewer suggestion.

  • Page 5, Line 207: I do not know what ‘manualised internally by the authors’ means – please use simpler language.

Simpler language has been used and the specific section revised to increase clarity.

  • Page 6, Table 1: This is a good table which outlines each intervention well. It would be good to more clearly differentiate between Sessions 1,2 and 3, either by including a line between them, or using shading.

A line has been included to increase clarity thanks to the reviewer comment

  1. RESULTS
  • Page 9, Figure 2: a) the last line in the allocation cell for ACT is missing. I assume it should include n=8?

Figure 2 has been deleted and newly updated figures containing all information have been inserted in the paper.

  • b) For the CBT allocation cell, the authors state that n=82 received intervention but 81 were allocated to intervention and 2 did not receive intervention (adds up to 83). Check numbers.

Numbers have been carefully revised according to the reviewer's comment.

  • Page 13, Lines 443-446: More than half of the participants continued to lose weight during the follow up period – more could be made of this in the discussion.

In line with the reviewer's suggestion, the discussion section has been entirely revised to better explain the results of the present paper.

  1. DISCUSSION
  • Overall, the focus of the discussion seems to be heavily on the scores of CORE-OM and AAQ-II which is all relevant (as long as it is not simply repeating the results). The implications of these findings needs to be more specifically discussed, with reference to the literature and as this is a novel study, borrowing from other areas where such interventions have been successfully implemented could be included.

Thanks to the reviewer's comment, the entire section has been revised by the authors. Implications of our findings are now more deeply discussed in the paper, with references to the on-topic literature and suggestions for clinical practice.

  • The discussion lacks mention of how these findings fit with current literature and thinking around weight loss maintenance interventions, and in particular how the behaviour change literature in primary care settings. It would also be of interest to readers to set this more in the context of how ACT and its focus assists participants to maintain weight loss once they are no longer in an in-patient setting. What is the real-world relevance of this research? And how do psychological interventions such as these fit within the context of lifestyle behaviour change interventions.

In line with reviewer comments, the entire section has been revised, integrated with more insights on our results and comparison with the current literature. The real-world relevance of the present study has been discussed and integrated into the discussion section as well according to the reviewer's suggestion.

  1. CONCLUSION
  • Page 15, Lines 519-522: This paragraph is a repeat of what is included in the introduction. Suggest simplifying and re-wording.

The entire section has been re-worded and simplified following the reviewer's suggestion.

Round 2

Reviewer 1 Report

Please provide a justification for why your trial was prospectively registered and include a section in your limitation section explaining what the impact of this may be.

Author Response

Response to reviewer comments for:

- Manuscript ID: ijerph-1305537

- Title: “ACTonFood. Acceptance and Commitment Therapy-based group treatment compared to Cognitive Behavioral Therapy-based group treatment for weight maintenance: an individually randomized group treatment trial”

- Journal: International Journal of Environmental Research and Public Health

We would like to thank the reviewers again for their valuable advice and help in improving our work as well as for the time dedicated to us, their interest in our study, and lastly for all of their meaningful suggestions.

Provided below is a point-by-point response describing our attempts to address all of their revisions in our manuscript.

REQUESTED REVISION

Reviewer  

  • Please provide a justification for why your trial was prospectively registered and include a section in your limitation section explaining what the impact of this may be.

Thank you for raising our attention on this topic. Thank you for raising our attention on this topic. According to your comments, we have added a section focusing on prospective registration and its meaning in our work. We also emended the text, carefully checking English for a deep edit.

Reviewer 2 Report

Thank you for your revisions to the manuscript which I believe have strengthened your paper in relation to the intro, methods and results sections. You have made some amendments to the discussion (reference to a single intervention). The discussion could still be improved and the lifestyle behaviour change context for weight loss maintenance further discussed, including the ongoing weight loss experienced after discharge by some participants. The fact that this study includes a 6-month follow up is a strength and with positive findings for ACT. More could be included about the challenges of real-world living and weight loss maintenance (with references to the current literature). 

Author Response

Response to reviewer comments for:

- Manuscript ID: ijerph-1305537

- Title: “ACTonFood. Acceptance and Commitment Therapy-based group treatment compared to Cognitive Behavioral Therapy-based group treatment for weight maintenance: an individually randomized group treatment trial”

- Journal: International Journal of Environmental Research and Public Health

We genuinely thank all reviewers for the time dedicated to revising our paper, their interest in our study, and all of their meaningful suggestions.

Provided below is a point-by-point response describing our attempts to address all of their revisions in our manuscript.

REQUESTED REVISION

Reviewer  

  • Thank you for your revisions to the manuscript which I believe have strengthened your paper in relation to the intro, methods and results sections. You have made some amendments to the discussion (reference to a single intervention). The discussion could still be improved and the lifestyle behaviour change context for weight loss maintenance further discussed, including the ongoing weight loss experienced after discharge by some participants. The fact that this study includes a 6-month follow up is a strength and with positive findings for ACT. More could be included about the challenges of real-world living and weight loss maintenance (with references to the current literature).

Thank you for your valuable comments that have significantly improved the overall paper. We have revised the discussion section further as requested.

More detailed pieces of information pointing out the potential contribution of ACT for lifestyle change have now been added to the text. We believe this revised version of our study could better fit the journal’s standard and aims, as well as readers' interests.
